# Partial Hepatic Vein Occlusion and Venous Congestion in Liver Exploration Using a Hyperspectral Camera: A Proposal for Monitoring Intraoperative Liver Perfusion

**DOI:** 10.3390/cancers15082397

**Published:** 2023-04-21

**Authors:** Simone Famularo, Elisa Bannone, Toby Collins, Elisa Reitano, Nariaki Okamoto, Kohei Mishima, Pietro Riva, Yu-Chieh Tsai, Richard Nkusi, Alexandre Hostettler, Jacques Marescaux, Eric Felli, Michele Diana

**Affiliations:** 1Department of Biomedical Sciences, Humanitas University, Pieve Emanuele, 20133 Milan, Italy; 2Department of Hepatobiliary and General Surgery, IRCCS Humanitas Research Hospital, Rozzano, 20089 Milan, Italy; 3Research Institute against Digestive Cancer (IRCAD), 67091 Strasbourg, France; 4Department of General Surgery, Poliambulanza Foundation Hospital, 25124 Brescia, Italy; 5Photonics Instrumentation for Health, iCube Laboratory, University of Strasbourg, 67000 Strasbourg, France; 6Department of General, Digestive and Endocrine Surgery, University Hospital of Strasbourg, 67000 Strasbourg, France; 7Department of Visceral Surgery and Medicine, Inselspital, Bern University Hospital, University of Bern, 3012 Bern, Switzerland

**Keywords:** hepatic vein occlusion, liver congestion, outflow occlusion, liver surgery, liver perfusion, hyperspectral camera

## Abstract

**Simple Summary:**

No measures of the changes occurring in the liver in the case of outflow deprivation during hepatectomies are available. This study investigated four indexes obtained using hyperspectral imaging in a pig model, measured in different liver lobes, before and after hepatic vein occlusion. The variations of StO_2_% and deoxy-Hb were considerably good markers of venous liver congestion.

**Abstract:**

Introduction. The changes occurring in the liver in cases of outflow deprivation have rarely been investigated, and no measurements of this phenomenon are available. This investigation explored outflow occlusion in a pig model using a hyperspectral camera. Methods. Six pigs were enrolled. The right hepatic vein was clamped for 30 min. The oxygen saturation (StO_2_%), deoxygenated hemoglobin level (de-Hb), near-infrared perfusion (NIR), and total hemoglobin index (THI) were investigated at different time points in four perfused lobes using a hyperspectral camera measuring light absorbance between 500 nm and 995 nm. Differences among lobes at different time points were estimated by mixed-effect linear regression. Results. StO_2_% decreased over time in the right lateral lobe (RLL, totally occluded) when compared to the left lateral (LLL, outflow preserved) and the right medial (RML, partially occluded) lobes (*p* < 0.05). De-Hb significantly increased after clamping in RLL when compared to RML and LLL (*p* < 0.05). RML was further analyzed considering the right portion (totally occluded) and the left portion of the lobe (with an autonomous draining vein). StO_2_% decreased and de-Hb increased more smoothly when compared to the totally occluded RLL (*p* < 0.05). Conclusions. The variations of StO_2_% and deoxy-Hb could be considered good markers of venous liver congestion.

## 1. Introduction

Several recent studies have shown that parenchymal-sparing hepatic resections for malignant liver disease can increase the likelihood of liver salvage in case of recurrence, boosting the number of patients who would benefit from liver resection [1]. These findings have led to an increased use of multiple nonanatomic resections (NAR) rather than major hepatectomies to achieve tumor clearance. However, parenchymal-sparing liver resection (PSR) requires a deep understanding of intraoperative navigation using ultrasonography (IOUS), particularly compared to simpler, conventional, major hepatectomies. During the latter, major vessels are identified and followed vertically, whereas during PSR, continuous employment of the IOUS is mandatory for identifying all terminal vascular branches that should be selectively resected or preserved [2]. These new procedures rely on the evidence of communicating veins among adjacent main hepatic veins (HV) [3,4], which can be used to resect the proximal part of any major HV without affecting the outflow drainage of distal liver tissues. However, switching to intraoperative liver navigation with IOUS raises new issues in liver surgery: the preservation of vital liver tissues becomes fundamental but more difficult to achieve. These could lead to an increased rate of remnant liver ischemia (RLI) [5], which can be caused either by remnant non-perfused liver tissue due to inappropriate inflow or by unintentional damage to tissue outflow. While inflow impairment has been thoroughly investigated, the changes occurring in the liver in cases of outflow deprivation have rarely been investigated, as at present, no available measurement of such an event has been reported. As a result, if the hepatic vein is occluded, what can be expected to happen to hepatic hemodynamics? This investigation explores outflow occlusion in a pig model using the study of the tissues’ absorbance in the visible and near-infrared spectra of light acquired by means of a hyperspectral camera.

## 2. Materials and Methods

This is a prospective observational non-randomized study in an animal model, conducted at the IRCAD Institute (Strasbourg, France) in October 2022. The aim of this study was to investigate the variations of four indexes (namely, oxygen saturation, StO_2_%; total hemoglobin index, THI; near-infrared perfusion index, NIR; total deoxy-hemoglobin, deoxy-Hb) among liver lobes with different outflow patency statuses, measured by means of a hyperspectral camera (details provided in the Methods section) at different time points. The considered time points were as follows: t0 = after laparotomy before hepatic vein occlusion; t1 = at the moment of hepatic vein occlusion; t2 = 10 min after occlusion; t3 = 30 min after occlusion. After t3, the vein was unclamped. T4 was 10 min after clamp removal.

To better understand the protocol, it can be useful to review the peculiarity of the anatomy of a pig’s liver. The liver is composed of five macroscopically independent lobes (right lateral lobe [RLL], right medial lobe [RML], left medial lobe [LML], left lateral lobe [LLL], and caudate lobe), each receiving its portal triads (inflow). Porcine hepatic veins are comprised of a left hepatic vein (LHV), a left middle hepatic vein (LMHV), a middle hepatic vein (MHV), a right middle hepatic vein (RMHV), and a right hepatic vein (RHV). The LHV, LMHV, and RHV drain the left lateral, left medial, and right lateral lobes, respectively. Only the right medial lobe is drained by two major hepatic veins, i.e., the MHV and RMHV. These two veins usually share a common trunk at the hepatocaval confluence. In order to conduct this experiment, the RML was purposely divided into two areas of vein drainage: the right side (RMLr) drained by the RMHV, and the left side (RMLl) drained by the MHV. No collateral veins between the hepatic veins have ever been described consistently with the macroscopic independence of the liver lobes. According to such peculiarities, the liver lobes investigated were the RLL (in which the outflow was totally occluded), the RML (both right and left sides, in which the RMLr drainage was occluded, and the RMLl was not), and the left lateral lobe (LLL, in which the outflow was always preserved and considered as the reference). Three different models were subsequently considered: the RLL model, in which the outflow was totally occluded without any possibility for alternative drainage; the RML model, in which the outflow was partially occluded and eventual interactions between regions drained by different main hepatic veins could occur; and, finally, the LLL model, in which the outflow was never interrupted and potential modifications could be related to intra-organ feedback.

### 2.1. Animals

The present study was part of the EXMachyna3 project (Intraoperative EXamination Using MAChine-learning-based HYperspectral for diagNosis & Autonomous Anatomy Assessment, Strasbourg, France), registered at ClinicalTrials.gov (NCT04589884) and approved by the local ethics committee of the School of Medicine at the University of Strasbourg (ID-RCB: 2020-A01896-33). All animals used in the experiment were managed in accordance with French regulations for animal use and care, and with the directives of the European Community Council (2010/63/EU) and ARRIVE guidelines [6]. Six adult male swine (Sus scrofa ssp. domesticus, mean weight: 32.4 ± 4.4 kg) were housed and acclimatized for 48 h in an enriched environment, respecting circadian cycles of light–darkness, with constant humidity and temperature conditions. They fasted 24 h before surgery, with ad libitum access to water, and were finally sedated (zolazepam + tiletamine 10 mg/kg IM) 30 min before the procedure to decrease stress. Anesthesia was performed intravenously (18-gauge IV catheter in-ear vein) with Propofol 3 mg/kg and maintained with rocuronium 0.8 mg/kg along with inhaled isoflurane 2% via the automatic standard respiratory system. Vital parameters were monitored through a mechanical ventilator machine and heartbeat was monitored with a pulse oximeter (Mindray PM-60). At the end of the procedure, all animals were euthanized with a lethal dose of pentobarbital (40 mg/kg).

### 2.2. Surgical Procedure

A midline laparotomy was performed to access the abdominal cavity. Xiphoidectomies were routinely performed to maximize liver visibility. Intraoperative ultrasounds were routinely performed to study hepatic vein anatomy and to exclude anatomical variations that could impact the experimental results. In particular, in all pigs, the RHMV and the RHV had a common confluence at the vena cava, and the MHV had an independent insertion. The confluence of the RHV and the MHRV was dissected and isolated from right to left extrahepatically, and a vascular clamp was positioned at the time of clamping (t1). The efficacy of the clamping was confirmed by means of intraoperative Doppler ultrasound. After 30 min of venous clamping, the vascular clamp was removed. At each time point, capillary lactates were analyzed by puncturing Glisson’s capsule on the back side of the RLL and measured using a strip-based portable lactate analyzer, which presents a margin of error of 0.35 mmol/L (EDGE, ApexBio, Taipei, Taiwan).

### 2.3. Hyperspectral Imaging

A CMOS push-broom scanning hyperspectral camera (TIVITA, Diaspective Vision GmbH, Am Salzhaff, Germany) was used to generate HS images, performed with a camera-specific software module from the same company. It enables the acquisition of optical remission spectroscopy in the visible and near-infrared range (400–1000 nm) as a contact-free method without any influence on the measured tissue. The tridimensional hypercube is composed of a spatial resolution (x, y) plus a third dimension with the relative reflectance of each pixel (z). The range of the wavelength detected is 500–1000 nm with a 5 nm interval, totaling 100 wavelengths for every pixel. The scanning method is passed through a slit-shaped aperture, which is motorized with an internal stepper motor [7]. The resolution of the hypercube is 640 × 480 pixels × 100 wavelengths. Image acquisition is performed at a distance of ~40 cm from the sample and monitored by a Bluefruit Feather nRF52832 distance sensor with an Adafruit VL53LOx device (Adafruit, New York, NY, USA) orthogonal to the liver’s surface. The light source is composed of six halogen lamps of 20 Watts (OSRAM Halospot 70, OSRAM GmbH, Munich, Germany). The HS camera takes about six seconds to perform the acquisition of the hypercube, which is transferred to a PC where it is processed, creating pseudo-color images. The relative reflectance:

(I/I0)

is converted into relative absorbance through the equation

A = −log(I/I0).

The device used in this experiment provides different algorithms (preset) that quantify the relative oxygen saturation (StO_2_%) of microcirculation at a depth of ~1 mm, and at deeper layers with the near-infrared (NIR) spectrum (3–5 mm) [8]. Briefly, StO_2_% is calculated with an algorithm based on the second derivative of the absorption spectra (570–590 nm and 740–780 nm). The NIR perfusion index is calculated with the absorbance spectra in a spectral range of 655–735 nm and 825–925 nm.

Another algorithm allowed us to estimate the total hemoglobin index (THI), which represents the sum of the oxygenated and deoxygenated hemoglobin in the tissue explored. Relatively deoxygenated hemoglobin (deoxy-Hb) was estimated as a consequence of the formula that enabled estimating the StO_2_% and the THI by subtracting the percentage of oxygenated hemoglobin from the THI value. Quantitative analysis of those four indexes was performed intraoperatively using the TIVITA Suite software module over four regions of interest (ROI), manually set in the RLL, RMLr, RMLr, and LLL. An example of the ROI positioning is depicted in Figure 1. The methods and algorithms of the TIVITA system were explained in more detail by Holmer et al. in 2018 [8].

### 2.4. Statistical Analysis

Data were reported as mean and standard deviation (SD). Normality was assessed with the Kolmogorov–Smirnov test at each time point. The variation of the four indexes among liver lobes was measured at each time point by fitting a linear mixed model (estimated using REML and nloptwrap optimizer). The models included the pigs’ IDs as a random effect (formula: ~1|ID). Standardized parameters were obtained by fitting the model on a standardized version of the dataset, and 95% confidence intervals (CIs) and *p* values were computed using a Wald t-distribution approximation. The same analysis was repeated after recording measurements in the LLL to account for the differences among the three occluded models. This latter analysis was repeated twice, first with RLL as the reference, and then with RMLl as a reference. All tests were two-tailed, and a *p* value < 0.05 was considered significant. The analyses were performed using R software (v. 4.2.2, libraries: lme4, ggplot2, report, pavo).

## 3. Results

An example of the photos acquired with the TIVITA system at each time point is depicted in Figure 2, showing the visible change on the liver’s surface after partial outflow occlusion.

The measured absorbance within the spectra among lobes at each time point was depicted in Appendix A. The trends of the indexes measured with the hyperspectral camera and the lactate are reported in Figure 3A–E at each time point. Each index was then analyzed to estimate its variation among the different lobes per each time point. Considering the lactate variation over time, no significant differences were measured, with an almost linear trend over time (beta = −0.04, 95% CI [−0.24, 0.16], t(96) = −0.39, *p* = 0.698; Std. beta = −0.03, 95% CI [−0.17, 0.11]).

### 3.1. StO_2_% Variations

The model’s total explanatory power for StO_2_% at t0 is substantial (conditional R2 = 0.85), and the part related to the fixed effects alone (marginal R2) is 9.37 × 10^−3^. The model’s intercept, corresponding to site = LLL, is at 38.33 (95% CI [22.61, 54.06], t(18) = 5.12, *p* < 0.001). No significant differences were observed between lobes.

At t1, the model’s total explanatory power is substantial (conditional R2 = 0.82), and the part related to the fixed effects alone (marginal R2) is 0.76. The model’s intercept is at 50.80 (95% CI [42.96, 58.64]). Within this model: (1) the effect of the site [RLL] is statistically significant and negative (beta = −31.40, 95% CI [−41.05, −21.75], t(14) = −6.98, *p* < 0.001, Std. beta = −1.93); (2) the effect of the site [RMLr] is statistically significant and negative (beta = −24.40, 95% CI [−34.05, −14.75], t(14) = −5.42, *p* < 0.001, Std. beta = −1.50).

At t2, the model’s total explanatory power is substantial (conditional R2 = 0.79), and the part related to the fixed effects alone (marginal R2) is 0.37. The model’s intercept is at 51.00 (95% CI [36.28, 65.72]). Within this model: (1) the effect of the site [RLL] is statistically significant and negative (beta = −30.67, 95% CI [−42.69, −18.65], t(18) = −5.36, *p* < 0.001, Std. beta = −1.49); (2) the effect of the site [RMLr] is statistically significant and negative (beta = −21.83, 95% CI [−33.85, −9.81], t(18) = −3.82, *p* = 0.001, Std. beta = −1.06). At t3, the model’s total explanatory power is substantial (conditional R2 = 0.84), and the part related to the fixed effects alone (marginal R2) is 0.52. The model’s intercept is at 56.02 (95% CI [41.17, 70.87]). Within this model: (1) the effect of the site [RLL] is statistically significant and negative (beta = −31.42, 95% CI [−43.86, −18.98], t(13) = −5.46, *p* < 0.001, Std. beta = −1.55); (2) the effect of the site [RMLr] is statistically significant and negative (beta = −22.42, 95% CI [−34.86, −9.98], t(13) = −3.89, *p* = 0.002, Std. beta = −1.11). At t4, the model’s total explanatory

power is substantial (conditional R2 = 0.79), and the part related to the fixed effects alone (marginal R2) is 0.05. The model’s intercept is at 57.25 (95% CI [39.54, 74.96]); no significant variations among lobes were identified. Results are summarized in Table 1 (random effects results in Appendix A).

Temporal trends were depicted in Figure 3A. Comparisons among occluded lobes after the exclusion of the LLL were reported in Table 2 (random effects results in Appendix A) when compared to RLL, while in Appendix A, they were compared to RMLl.

### 3.2. Deoxy-Hb Variations

At t0, the model’s total explanatory power is substantial (conditional R2 = 0.77), and the part

related to the fixed effects alone (marginal R2) is 8.92 × 10^−3^. The model’s intercept,

corresponding to site = LLL, is at 40.87 (95% CI [27.03, 54.71], t(18) = 6.20, *p* < 0.001). No significant variations were observed among lobes at this time point.

At t1, the model’s total explanatory power is substantial (conditional R2 = 0.77), and the part related to the fixed effects alone (marginal R2) is 8.92e−03. The model’s intercept is at 40.87 (95% CI [27.03, 54.71]). Within this model, none of the lobes reached significance. At t2, the model’s total explanatory power is substantial (conditional R2 = 0.92), and the part related to the fixed effects alone (marginal R2) is 0.40. The model’s intercept is at 26.72 (95% CI [16.08, 37.37]). Within this model: (1) the effect of the site [RLL] is statistically significant and positive (beta = 24.14, 95% CI [18.73, 29.55], t(18) = 9.37, *p* < 0.001, Std. beta = 1.57); (2) the effect of the site [RMLr] is statistically significant and positive (beta = 10.35, 95% CI [4.94, 15.76], t(18) = 4.02, *p* < 0.001,Std. beta = 0.67). At t3, the model’s total explanatory power is substantial (conditional R2 = 0.81), and the part related to the fixed effects alone (marginal R2) is 0.40. The model’s intercept is at 21.11 (95% CI [7.03, 35.19]). Within this model: (1) the effect of the site [RLL] is statistically significant and positive (beta = 27.62, 95% CI [16.47, 38.76], t(14) = 5.32, *p* < 0.001, Std. beta = 1.53); (2) the effect of the site [RMLr] is statistically significant and positive (beta = 17.00, 95% CI [5.86, 28.15], t(14) = 3.27, *p* = 0.006, Std. beta = 0.94). At t4, the model’s total explanatory power is substantial (conditional R2 = 0.81), and the part related to the fixed effects alone (marginal R2) is 0.07. The model’s intercept is at 24.81 (95% CI [9.31, 40.32]), and no significant variations among lobes were detected. Results are summarized in Table 1 (random effects results in Appendix A). Temporal trends were depicted in Figure 3B. Comparisons among occluded lobes after the exclusion of the LLL were reported in Table 2 (random effects results in Appendix A) when compared to RLL, and in Appendix A when compared to RMLl.

### 3.3. NIR Variations

At t0, the model’s total explanatory power is substantial (conditional R2 = 0.32), and the part

related to the fixed effects alone (marginal R2) is 3.52e−03. The model’s intercept,

corresponding to site = LLL, is at 58.33 (95% CI [47.56, 69.11], t(18) = 11.37, *p* < 0.001). No significant variations were observed among lobes at this time point. At t1, the model’s total explanatory power is substantial (conditional R2 = 0.35), and the part related to the fixed effects alone (marginal R2) is 0.35. The model’s intercept is at 53.60 (95% CI [40.10, 67.10]). Within this model, the effect of the site [RLL] is statistically significant and negative (beta = −25.00, 95% CI [−44.01, −5.99], t(14) = −2.82, *p* = 0.014, Std. beta = −1.51).

At t2, the model’s total explanatory power is substantial (conditional R2 = 0.35), and the part related to the fixed effects alone (marginal R2) is 0.35. The model’s intercept is at 53.60 (95% CI [40.10, 67.10]). Within this model, the effect of the site [RLL] is statistically significant and negative (beta = −25.00, 95% CI [−44.01, −5.99], t(14) = −2.82, *p* = 0.014, Std. beta = −1.51). At t3, the model’s total explanatory power is substantial (conditional R2 = 0.33), and the part related to the fixed effects alone (marginal R2) is 0.13. The model’s intercept is at 52.44 (95% CI [35.10, 69.78]). Within this model, no significant variations between lobes were observed. At t4, the model’s total explanatory power is substantial (conditional R2 = 0.80), and the part related to the fixed effects alone (marginal R2) is 0.09. The model’s intercept is at 65.75 (95% CI [55.87, 75.63]). Within this model, the effect of the site [RLL] is statistically significant and negative (beta = −6.75, 95% CI [−13.30, −0.20], t(10) = −2.30, *p* = 0.045, Std. beta = −0.80). Results are summarized in Table 1 (random effects results in Appendix A). Temporal trends were depicted in Figure 3C. Comparisons among occluded lobes after the exclusion of the LLL were reported in Table 2 (random effects results in Appendix A) when compared to RLL, and in Appendix A for RMLl.

### 3.4. THI Variations

At t0, the model’s total explanatory power is substantial (conditional R2 = 0.59), and the part

related to the fixed effects alone (marginal R2) is 0.03. The model’s intercept,

corresponding to site = LLL, is at 65.17 (95% CI [54.64, 75.69], t(18) = 13.01, *p* < 0.001). No significant variations were observed among lobes at this time point.

At t1, the model’s total explanatory power is substantial (conditional R2 = 0.59), and the part related to the fixed effects alone (marginal R2) is 0.03. The model’s intercept is at 65.17 (95% CI [54.64, 75.69]). Within this model, the variations in the index between lobes were not significant. At t2, the model’s total explanatory power is substantial (conditional R2 = 0.66), and the part related to the fixed effects alone (marginal R2) is 0.16. The model’s intercept is at 52.83 (95% CI [42.56, 63.11]). Within this model, the effect of the site [RLL] is statistically significant and positive (beta = 10.67, 95% CI [1.38, 19.95], t(18) = 2.41, *p* = 0.027, Std. beta = 0.86). At t3, the model’s total explanatory power is substantial (conditional R2 = 0.77), and the part related to the fixed effects alone (marginal R2) is 0.15. The model’s intercept is at 55.54 (95% CI [43.68, 67.41]). At t4, the model’s total explanatory power is substantial (conditional R2 = 0.74), and the part related to the fixed effects alone (marginal R2) is 0.14. The model’s intercept is at 53.75 (95% CI [41.08, 66.42]), but no significance has been detected. Results are summarized in Table 1 (random effects results in Appendix A). Temporal trends are depicted in Figure 3D. Comparisons among occluded lobes after the exclusion of the LLL are reported in Table 2 (random effects results in Appendix A) when compared to RLL, and in Appendix A when compared to RMLl.

## 4. Discussion

The lobes impacted by HV occlusion showed significant variations in the hemodynamics parameter measured by our study when compared to those with preserved outflow. In particular, StO_2_% decreased significantly at each time point after occlusion in the RLL and in the RMLr when compared to LLL, returning to normal after outflow restoration, while deoxy-Hb showed the opposite behavior, with a significant increase over time. The NIR index, a parameter estimating perfusion deeper than StO_2_%, decreased after venous occlusion in the RLL and RMLr, but remained significantly decreased only in RLL at the other time points, returning to normal after declamping. These variations were probably the direct indication of liver congestion; when outflow is occluded, the amount of deoxygenated hemoglobin increased, as a consequence of the impossibility of effectively draining venous blood.

It is of interest that the lactates did not vary during time points, and they were not related to the hemodynamic indexes; although this parameter is the most widely used in the liver surgical field to monitor circulatory impairment, with an evident predictor role in case of inflow occlusion [9], they seemed to be useful in cases of outflow occlusion. Consequently, their role in estimating venous congestion was not validated by our measurements, demonstrating that the hyperspectral indexes can provide more robust evidence of the changes occurring in liver tissue.

When considering the temporal trends, other crucial information can be derived from our study. The indexes in the LLL were always very stable over time, and a similar and superimposable behavior was also recognized on the left side of the RML, where outflow was ensured by the MHV. Notwithstanding, the counterpart of the RML, the right side, showed trends closer to the totally excluded lobe (the RLL). The RML, although fed by two hepatic veins, is macroscopically a single structure, where indeed connections between veins were not recorded on the IOUS before and during clamping. However, the variations of the indexes over time were less sharp than those observed in the RLL. When compared to the variations in RMLl, a more contained reduction in deoxy-Hb and StO_2_% was observed compared to that observed in the RLL. As a confirmation, when the RMLl was taken as the reference, in RMLr, StO_2_% significantly increased and deoxy-Hb decreased over time during the occlusion period compared to the trends in the RLL. This observation can be attributed to the presence of venous communications among the clamped RMHV and the MHV that, even if not visualized on the IOUS, became patent after outflow occlusion to create a sort of compensation for the outflow pathway. This phenomenon was probably not observed in the RLL due to the anatomic separation between the RLL and the RML. Recently, the presence and the role of communicating veins (CV) between adjacent HVs were described in surgical human settings [10,11]; these veins were recognized in up to 80% of livers intraoperatively, even if they could not always be visualized directly via IOUS due to their small size [4]. More importantly, in autopsy specimens, it has been demonstrated that these veins were present in almost all cases [12]. It has also been reported that, when CVs are not visualizable, flow assessment using color Doppler US or e-flow US can confirm their presence by the absence of portal flow inversion (from hepatopetal to hepatofugal) [2]. Again, this change of direction in the portal flow is a rare finding. The physiological changes behind flow inversion rely on the evidence that, when the outflow is occluded, the portal vein seems to become the outflow pathway in the absence of CVs, while the arterial flow is increased, becoming the main inflow route [13,14]. In cases when no CVs are present, in our RML model, one can expect to observe different variations in the hemodynamic indexes of the RMLr, with rebound feedback in the RMLl. However, our measurements showed that the behavior of the RMLl was almost stable over time, exactly as the LLL, which was never occluded. Indeed, index variations were observed only in the RMLr, where the reduction of oxygenated blood and the increase of deoxygenated blood were slightly less significant when compared to the total excluded lobe, and, in general, more rapid in restoring the original physiological condition. We hypothesized that these findings suggested the presence and the opening of CVs among the clamped RMHV and the patent MHV. Accordingly, hyperspectral estimation of those indexes could be a novel method to estimate the presence of CVs in an intraoperative setting. However, this conclusion should be carefully considered, as it is mostly speculative, and other studies in human settings during clamping using color Doppler or e-flow IOUS should still be performed.

Potential surgical applications of the hyperspectral camera’s indexes have already been investigated [15,16,17], but another valuable application of our method could be the intraoperative estimation of remnant liver congestion. Recently, it has been reported that remnant liver ischemia should be carefully considered in human oncological settings because of a significantly increased risk of morbidity, recurrence, and mortality [5]. Such a phenomenon may occur not only in cases of altered portal vascularization of a liver segment but also in cases of preservation of liver tissue without appropriate outflow drainage. At present, there are no means to estimate appropriate outflow preservation of the remnant liver intraoperatively, and the finding is always recognized during postoperative CT scans. In our totally occluded model (the RLL), the trends among the times of StO_2_% and deoxy-Hb showed a clear trend that could be applied in operating rooms to estimate congestion. Consequently, our method could provide comparative assessment between different sites of the remnant liver, as well as with the liver prior to transection. This application can be performed in real time, as it is not time-consuming. Moreover, if our measurements are confirmed, the hyperspectral camera could be considered to estimate intraoperatively the remnant liver ischemia (RLI) [5], which has been recently associated with an increased rate of postoperative complications and worse long-term results.

The present study has several limitations. Fundamentally, it is an observational study, without any comparison of clinical effects. This means that the clinical implications put forward should be carefully evaluated in further studies, and they should be considered speculative at this time. In addition, remission spectrometry in in vivo tissues cannot provide a real quantification of the concentration of hemoglobin as expected; e.g., in the case of a biochemical analysis. This means that the values presented here (a part of SatO2%) are absolute values without a specific unit of measure. Another limitation is that, regarding the use of the indexes as markers of liver venous congestion, no cut-offs were provided in this study because of its design; even in this case, other studies should be conducted to further clarify the details of the potential clinical applications.

## 5. Conclusions

The present study opens the door to a novel application of the hyperspectral camera in liver surgery as an instrument to detect the presence of vascular outflow variations during hepatectomies. In the near future, other studies in the human setting should be performed to assess the correlation between such indexes and the clinical events reported in the case of liver outflow congestion after surgery.

## Figures and Tables

**Figure 1 cancers-15-02397-f001:**
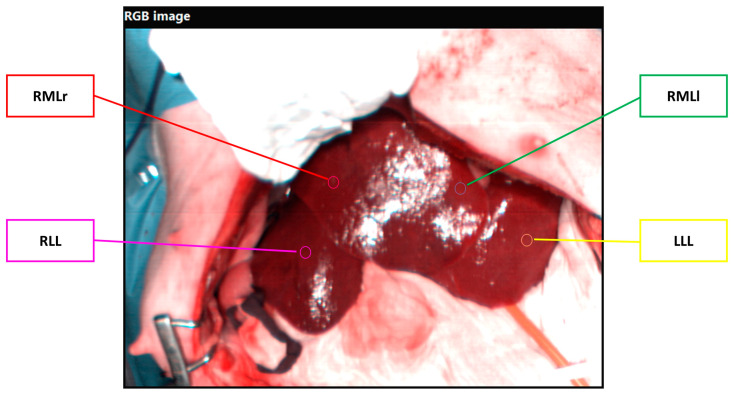
Example of the ROI positioning to extract the indexes.

**Figure 2 cancers-15-02397-f002:**
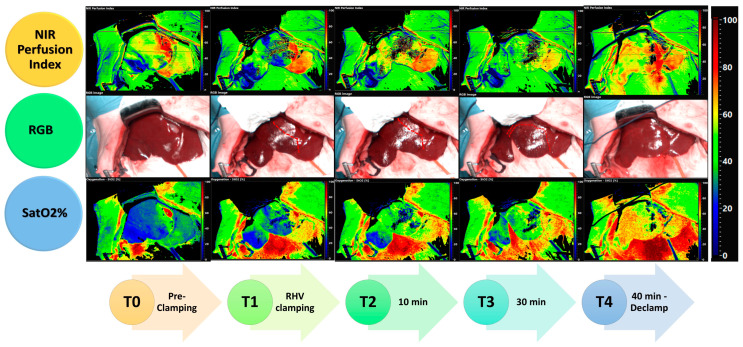
Examples of the photos acquired with the hyperspectral camera at different time points. In the first line, the near-infrared perfusion index window was controlled; in the second, the real appearance of the pig liver at different time points; in the third line, the window to check the oxygen saturation is evident.

**Figure 3 cancers-15-02397-f003:**
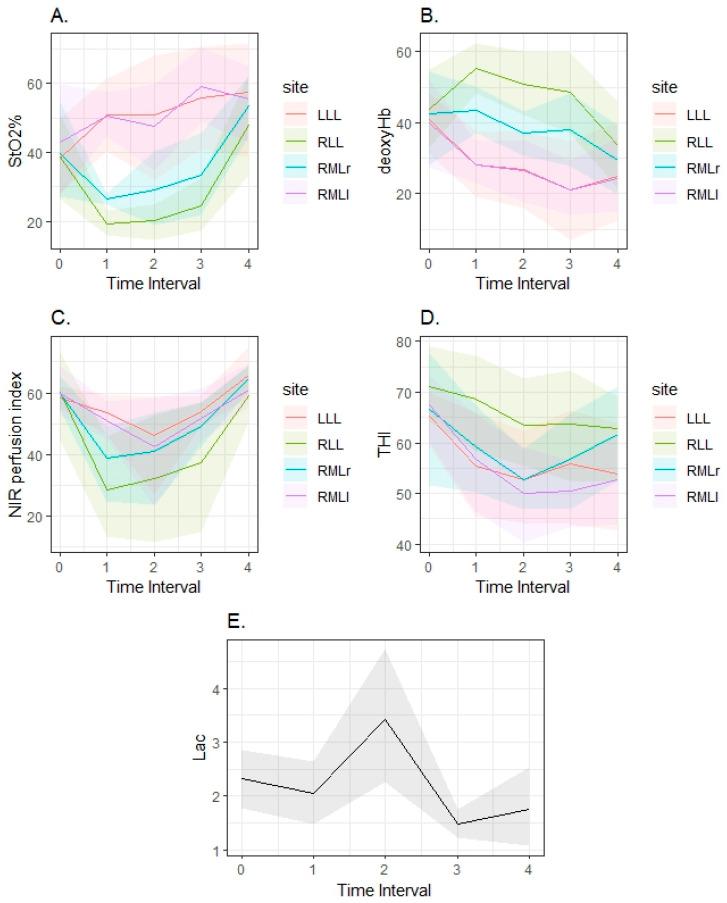
Trends of each index during time in the different liver lobes. (**A**) Oxygen saturation, (**B**) deoxygenated hemoglobin, (**C**) near-infrared perfusion index, (**D**) total hemoglobin index, (**E**) lactates.

**Table 1 cancers-15-02397-t001:** Mixed-effects linear regression to estimate the variations of the hyperspectral indexes among different liver lobes. In this table, the reference level is the left lateral lobe.

T	Index	Site	Coefficient	SE	95% CI	t	df	*p*
0	StO_2_%	(Intercept)	38.33	7.49	[22.61, 54.06]	5.12	18	<0.001
		RLL	0.5	4.14	[−8.20, 9.20]	0.12	18	0.905
		RMLr	1.5	4.14	[−7.20, 10.20]	0.36	18	0.721
		RMLl	4.5	4.14	[−4.20, 13.20]	1.09	18	0.292
	NIR	(Intercept)	58.33	5.13	[47.56, 69.11]	11.37	18	<0.001
		RLL	1.83	6	[−10.78, 14.45]	0.31	18	0.764
		RMLr	1.67	6	[−10.95, 14.28]	0.28	18	0.785
		RMLl	1.5	6	[−11.12, 14.12]	0.25	18	0.806
	THI	(Intercept)	65.17	5.01	[54.64, 75.69]	13.01	18	<0.001
		RLL	6	4.61	[−3.69, 15.69]	1.3	18	0.21
		RMLr	1.5	4.61	[−8.19, 11.19]	0.33	18	0.749
		RMLl	2.33	4.61	[−7.36, 12.03]	0.51	18	0.619
	DeHb	(Intercept)	40.87	6.59	[27.03, 54.71]	6.2	18	<0.001
		RLL	3.08	4.48	[−6.34, 12.49]	0.69	18	0.501
		RMLr	1.53	4.48	[−7.88, 10.95]	0.34	18	0.736
		RMLl	−0.85	4.48	[−10.26, 8.57]	−0.19	18	0.852
1	StO_2_%	(Intercept)	50.8	3.65	[42.96, 58.64]	13.9	14	<0.001
		RLL	−31.4	4.5	[−41.05, −21.75]	−6.98	14	<0.001
		RMLr	−24.4	4.5	[−34.05, −14.75]	−5.42	14	<0.001
		RMLl	−0.4	4.5	[−10.05, 9.25]	−0.09	14	0.93
	NIR	(Intercept)	53.6	6.3	[40.10, 67.10]	8.51	14	<0.001
		RLL	−25	8.86	[−44.01, −5.99]	−2.82	14	0.014
		RMLr	−14.8	8.86	[−33.81, 4.21]	−1.67	14	0.117
		RMLl	−2.6	8.86	[−21.61, 16.41]	−0.29	14	0.774
	THI	(Intercept)	55.4	5.55	[43.49, 67.31]	9.98	14	<0.001
		RLL	13.2	2.98	[6.81, 19.59]	4.43	14	<0.001
		RMLr	3.8	2.98	[−2.59, 10.19]	1.28	14	0.223
		RMLl	1.4	2.98	[−4.99, 7.79]	0.47	14	0.645
	DeHb	(Intercept)	28.16	4.44	[18.64, 37.68]	6.34	14	<0.001
		RLL	27.03	3.13	[20.31, 33.75]	8.63	14	<0.001
		RMLr	15.38	3.13	[8.66, 22.10]	4.91	14	<0.001
		RMLl	−0.06	3.13	[−6.78, 6.66]	−0.02	14	0.985
2	StO_2_%	(Intercept)	51	7	[36.28, 65.72]	7.28	18	<0.001
		RLL	−30.67	5.72	[−42.69, −18.65]	−5.36	18	<0.001
		RMLr	−21.83	5.72	[−33.85, −9.81]	−3.82	18	0.001
		RMLl	−3.33	5.72	[−15.35, 8.69]	−0.58	18	0.567
	NIR	(Intercept)	46.17	10.03	[25.09, 67.24]	4.6	18	<0.001
		RLL	−14	7.49	[−29.73, 1.73]	−1.87	18	0.078
		RMLr	−5.17	7.49	[−20.89, 10.56]	−0.69	18	0.499
		RMLl	−3.5	7.49	[−19.23, 12.23]	−0.47	18	0.646
	THI	(Intercept)	52.83	4.89	[42.56, 63.11]	10.8	18	<0.001
		RLL	10.67	4.42	[1.38, 19.95]	2.41	18	0.027
		RMLr	5.33E−15	4.42	[−9.29, 9.29]	1.21E−15	18	>0.999
		RMLl	−2.83	4.42	[−12.12, 6.45]	−0.64	18	0.53
	DeHb	(Intercept)	26.72	5.07	[16.08, 37.37]	5.27	18	<0.001
		RLL	24.14	2.58	[18.73, 29.55]	9.37	18	<0.001
		RMLr	10.35	2.58	[4.94, 15.76]	4.02	18	<0.001
		RMLl	−0.3	2.58	[−5.71, 5.12]	−0.11	18	0.91
3	StO_2_%	(Intercept)	56.02	6.87	[41.17, 70.87]	8.15	13	<0.001
		RLL	−31.42	5.76	[−43.86, −18.98]	−5.46	13	<0.001
		RMLr	−22.42	5.76	[−34.86, −9.98]	−3.89	13	0.002
		RMLl	2.98	5.76	[−9.46, 15.42]	0.52	13	0.614
	NIR	(Intercept)	52.44	8.03	[35.10, 69.78]	6.53	13	<0.001
		RLL	−15.04	9.62	[−35.83, 5.75]	−1.56	13	0.142
		RMLr	−3.44	9.62	[−24.23, 17.35]	−0.36	13	0.726
		RMLl	−0.64	9.62	[−21.43, 20.15]	−0.07	13	0.948
	THI	(Intercept)	55.54	5.49	[43.68, 67.41]	10.11	13	<0.001
		RLL	8.06	4.16	[−0.93, 17.05]	1.94	13	0.075
		RMLr	1.26	4.16	[−7.73, 10.25]	0.3	13	0.767
		RMLl	−5.14	4.16	[−14.13, 3.85]	−1.24	13	0.239
	DeHb	(Intercept)	21.11	6.56	[7.03, 35.19]	3.22	14	0.006
		RLL	27.62	5.2	[16.47, 38.76]	5.32	14	<0.001
		RMLr	17	5.2	[5.86, 28.15]	3.27	14	0.006
		RMLl	−0.1	5.2	[−11.24, 11.05]	−0.02	14	0.985
4	StO_2_%	(Intercept)	57.25	7.95	[39.54, 74.96]	7.2	10	<0.001
		RLL	−9.25	5.34	[−21.15, 2.65]	−1.73	10	0.114
		RMLr	−3.75	5.34	[−15.65, 8.15]	−0.7	10	0.499
		RMLl	−1.75	5.34	[−13.65, 10.15]	−0.33	10	0.75
	NIR	(Intercept)	65.75	4.43	[55.87, 75.63]	14.83	10	<0.001
		RLL	−6.75	2.94	[−13.30, −0.20]	−2.3	10	0.045
		RMLr	−1.25	2.94	[−7.80, 5.30]	−0.43	10	0.68
		RMLl	−4.75	2.94	[−11.30, 1.80]	−1.62	10	0.137
	THI	(Intercept)	53.75	5.68	[41.08, 66.42]	9.46	10	<0.001
		RLL	9	4.47	[−0.96, 18.96]	2.01	10	0.072
		RMLr	8	4.47	[−1.96, 17.96]	1.79	10	0.104
		RMLl	−1	4.47	[−10.96, 8.96]	−0.22	10	0.827
	DeHb	(Intercept)	24.81	6.96	[9.31, 40.32]	3.57	10	0.005
		RLL	8.85	4.42	[−1.00, 18.71]	2	10	0.073
		RMLr	4.52	4.42	[−5.33, 14.38]	1.02	10	0.331
		RMLl	−0.45	4.42	[−10.30, 9.41]	−0.1	10	0.921

**Table 2 cancers-15-02397-t002:** Mixed-effects linear regression to estimate the variations of the hyperspectral indexes among different liver lobes. In this table, the reference level is the right lateral lobe after the exclusion of the left lateral lobe.

T	INDEX	SITE	Coefficient	SE	95% CI	t	df	*p*
**0**	**StO_2_%**	**(Intercept)**	38.83	7.86	[21.85, 55.81]	4.94	13	<0.001
		**RMLr**	1	4.14	[−7.95, 9.95]	0.24	13	0.813
		**RMLl**	4	4.14	[−4.95, 12.95]	0.97	13	0.352
	**NIR**	**(Intercept)**	60.17	5.81	[47.62, 72.71]	10.36	13	<0.001
		**RMLr**	−0.17	6.95	[−15.18, 14.84]	−0.02	13	0.981
		**RMLl**	−0.33	6.95	[−15.34, 14.68]	−0.05	13	0.962
	**THI**	**(Intercept)**	71.17	5.48	[59.33, 83.00]	12.99	13	<0.001
		**RMLr**	−4.5	5.51	[−16.40, 7.40]	−0.82	13	0.429
		**RMLl**	−3.67	5.51	[−15.56, 8.23]	−0.67	13	0.517
	**DeHb**	**(Intercept)**	43.94	7.01	[28.79, 59.10]	6.26	13	<0.001
		**RMLr**	−1.54	4.97	[−12.27, 9.19]	−0.31	13	0.761
		**RMLl**	−3.92	4.97	[−14.65, 6.81]	−0.79	13	0.444
**1**	**StO_2_%**	**(Intercept)**	19.4	2.47	[13.90, 24.90]	7.86	10	<0.001
		**RMLr**	7	2.71	[0.95, 13.05]	2.58	10	**0.027**
		**RMLl**	31	2.71	[24.95, 37.05]	11.42	10	**<0.001**
	**NIR**	**(Intercept)**	28.6	6.94	[13.15, 44.05]	4.12	10	0.002
		**RMLr**	10.2	8.3	[−8.30, 28.70]	1.23	10	0.247
		**RMLl**	22.4	8.3	[3.90, 40.90]	2.7	10	**0.022**
	**THI**	**(Intercept)**	68.6	5.44	[56.49, 80.71]	12.62	10	<0.001
		**RMLr**	−9.4	2.36	[−14.67, −4.13]	−3.98	10	**0.003**
		**RMLl**	−11.8	2.36	[−17.07, −6.53]	−4.99	10	**<0.001**
	**DeHb**	**(Intercept)**	55.19	3.9	[46.49, 63.88]	14.14	10	<0.001
		**RMLr**	−11.65	3.07	[−18.50, −4.80]	−3.79	10	**0.004**
		**RMLl**	−27.09	3.07	[−33.94, −20.24]	−8.81	10	**<0.001**
**2**	**StO_2_%**	**(Intercept)**	20.33	5.71	[7.99, 32.68]	3.56	13	0.003
		**RMLr**	8.83	5.14	[−2.28, 19.94]	1.72	13	0.11
		**RMLl**	27.33	5.14	[16.22, 38.44]	5.32	13	**<0.001**
	**NIR**	**(Intercept)**	32.17	10.2	[10.13, 54.21]	3.15	13	0.008
		**RMLr**	8.83	8.4	[−9.32, 26.98]	1.05	13	0.312
		**RMLl**	10.5	8.4	[−7.65, 28.65]	1.25	13	0.233
	**THI**	**(Intercept)**	63.5	4.73	[53.27, 73.73]	13.42	13	<0.001
		**RMLr**	−10.67	4.5	[−20.39, −0.94]	−2.37	13	**0.034**
		**RMLl**	−13.5	4.5	[−23.22, −3.78]	−3	13	**0.01**
	**DeHb**	**(Intercept)**	50.86	4.58	[40.96, 60.76]	11.1	13	<0.001
		**RMLr**	−13.79	2.44	[−19.07, −8.52]	−5.65	13	**<0.001**
		**RMLl**	−24.44	2.44	[−29.71, −19.16]	−10.01	13	**<0.001**
**3**	**StO_2_%**	**(Intercept)**	24.6	6.21	[10.76, 38.44]	3.96	10	0.003
		**RMLr**	9	5.91	[−4.16, 22.16]	1.52	10	0.159
		**RMLl**	34.4	5.91	[21.24, 47.56]	5.82	10	**<0.001**
	**NIR**	**(Intercept)**	37.4	7.98	[19.61, 55.19]	4.68	10	<0.001
		**RMLr**	11.6	8.85	[−8.11, 31.31]	1.31	10	0.219
		**RMLl**	14.4	8.85	[−5.31, 34.11]	1.63	10	0.135
	**THI**	**(Intercept)**	63.6	5.11	[52.21, 74.99]	12.44	10	<0.001
		**RMLr**	−6.8	4.27	[−16.31, 2.71]	−1.59	10	0.142
		**RMLl**	−13.2	4.27	[−22.71, −3.69]	−3.09	10	**0.011**
	**DeHb**	**(Intercept)**	48.72	5.86	[35.66, 61.78]	8.31	10	<0.001
		**RMLr**	−10.61	4.56	[−20.78, −0.45]	−2.33	10	**0.042**
		**RMLl**	−27.72	4.56	[−37.88, −17.55]	−6.08	10	**<0.001**
**4**	**StO_2_%**	**(Intercept)**	48	7.07	[31.27, 64.73]	6.78	7	<0.001
		**RMLr**	5.5	5.49	[−7.48, 18.48]	1	7	0.35
		**RMLl**	7.5	5.49	[−5.48, 20.48]	1.37	7	0.214
	**NIR**	**(Intercept)**	59	4.4	[48.60, 69.40]	13.41	7	<0.001
		**RMLr**	5.5	3.15	[−1.96, 12.96]	1.74	7	0.125
		**RMLl**	2	3.15	[−5.46, 9.46]	0.63	7	0.546
	**THI**	**(Intercept)**	62.75	5.47	[49.82, 75.68]	11.48	7	<0.001
		**RMLr**	−1	4.86	[−12.49, 10.49]	−0.21	7	0.843
		**RMLl**	−10	4.86	[−21.49, 1.49]	−2.06	7	0.079
	**DeHb**	**(Intercept)**	33.67	6.29	[18.79, 48.55]	5.35	7	0.001
		**RMLr**	−4.33	4.33	[−14.56, 5.90]	−1	7	0.35
		**RMLl**	−9.3	4.33	[−19.53, 0.93]	−2.15	7	0.069

## Data Availability

Data will be accessible upon reasonable request to the corresponding author.

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
