# Peer review of "Partial Hepatic Vein Occlusion and Venous Congestion in Liver Exploration Using a Hyperspectral Camera: A Proposal for Monitoring Intraoperative Liver Perfusion"

_cancers, 2023, doi:10.3390/cancers15082397_

Round 1
Reviewer 1 Report
The article of dr. Famularo and col. is very interesting and approach a subject of concern in surgical practice from a innovative, novel perspective. The article describe an animal experimental model to investigate the effects upon liver in case of outflow deprivation. They found that variations of StO2% and deoxi-Hb could be promising biomarkers for liver venous congestion. The article is clearly written and aid significant value to the current knowledge on this topic.
For these reasons, I recommend accept for publication
Author Response
We really thank the reviewer for his/her appreciation of our study.
Reviewer 2 Report
It is an excellent well designed and well written study. Though experimental, the methods are well presented.
I have only one comment to make
1. In the section of discussion the authors must describe better which will be the usefulness of this new device in the clinical practice. Furthermore, they must highlight more clearly the impact of their results on our decisions in patients undergoing an hepatectomy
Author Response
Thank you very much for your eminent opinion on our paper.
We modified the discussion according to your indication.